# Epidemiology of Pneumococcal Pneumonia in Louisville, Kentucky, and Its Estimated Burden of Disease in the United States

**DOI:** 10.3390/microorganisms11112813

**Published:** 2023-11-20

**Authors:** Julio Ramirez, Stephen Furmanek, Thomas R. Chandler, Timothy Wiemken, Paula Peyrani, Forest Arnold, William Mattingly, Ashley Wilde, Jose Bordon, Rafael Fernandez-Botran, Ruth Carrico, Rodrigo Cavallazzi, The University of Louisville Pneumonia Study Group

**Affiliations:** 1Norton Infectious Diseases Institute, Norton Healthcare, Louisville, KY 40202, USA; 2School of Medicine, University of Louisville, Louisville, KY 40290, USA; 3Washington Health Institute, Washington, DC 20017, USA

**Keywords:** *Streptococcus pneumoniae*, community-acquired pneumonia, epidemiology, urinary antigen

## Abstract

*Streptococcus pneumoniae* remains a primary pathogen in hospitalized patients with community-acquired pneumonia (CAP). The objective of this study was to define the epidemiology of pneumococcal pneumonia in Louisville, Kentucky, and to estimate the burden of pneumococcal pneumonia in the United States (US). This study was nested in a prospective population-based cohort study of all adult residents in Louisville, Kentucky, who were hospitalized with CAP from 1 June 2014 to 31 May 2016. In hospitalized patients with CAP, urinary antigen detection of 24 *S. pneumoniae* serotypes (UAD-24) was performed. The annual population-based pneumococcal pneumonia incidence was calculated. The distribution of *S. pneumoniae* serotypes was characterized. Ecological associations between pneumococcal pneumonia and income level, race, and age were defined. Mortality was evaluated during hospitalization and at 30 days, 6 months, and 1 year after hospitalization. Among the 5402 CAP patients with a UAD-24 test performed, 708 (13%) patients had pneumococcal pneumonia. The annual cumulative incidence was 93 pneumococcal pneumonia hospitalizations per 100,000 adults (95% CI = 91–95), corresponding to an estimated 226,696 annual pneumococcal pneumonia hospitalizations in the US. The most frequent serotypes were 19A (12%), 3 (11%), and 22F (11%). Clusters of cases were found in areas with low incomes and a higher proportion of Black or African American population. Pneumococcal pneumonia mortality was 3.7% during hospitalization, 8.2% at 30 days, 17.6% at 6 months, and 25.4% at 1 year after hospitalization. The burden of pneumococcal pneumonia in the US remains significant, with an estimate of more than 225,000 adults hospitalized annually, and approximately 1 out of 4 hospitalized adult patients dies within 1 year after hospitalization.

## 1. Introduction

The epidemiology of *Streptococcus pneumoniae* community-acquired pneumonia (CAP) in adults varies across countries. This may be due to different rates of pneumococcal and influenza vaccination in adults, different patterns of cigarette smoking, and different implementation of pneumococcal vaccination in children [1,2,3,4,5,6,7,8]. One challenge in defining the burden of adult hospitalizations due to pneumococcal pneumonia is that the proportion of *S. pneumoniae* cases depends on the diagnostic techniques used for identification. The conventional microbiological approach for identifying patients with pneumococcal pneumonia typically involves analyzing sputum or other respiratory samples for Gram stain and culture, culturing blood or other sterile sites such as pleural fluid, and detecting *S. pneumoniae* polysaccharide C by using a urinary antigen.

In a recent CDC study, the proportion of hospitalized patients with pneumococcal pneumonia was 5% using standard microbiological workup [9]. When a urinary antigen detection assay for 13 *S. pneumoniae* serotypes was used in the same study population, the proportion of *S. pneumoniae* cases increased to 9% [10]. A urinary antigen detection immunodiagnostic assay was recently clinically validated to diagnose 11 additional serotypes, expanding the possibility of detecting 24 *S. pneumoniae* serotypes [11,12]. Only a limited number of studies have utilized the expanded urinary antigen detection (UAD-24) technology for detecting 24 *S. pneumoniae* serotypes as a diagnostic tool to determine the incidence of *S. pneumoniae* in hospitalized CAP patients. For instance, a Korean study involving 2669 hospitalized adults with CAP found that 9.4% of the cases were attributed to pneumococcal CAP [5]. An Italian study focusing on 1155 hospitalized elderly patients with CAP identified 13.1% of the cases as pneumococcal CAP [6]. A Swedish study involving 518 hospitalized adults with CAP revealed that 24.3% of the cases were pneumococcal CAP [7]. Lastly, a study conducted in Spain, which assessed 3107 hospitalized adults with CAP, identified 28.8% of the cases as pneumococcal CAP [8]. Importantly, all these studies highlighted that a significant proportion of patients with pneumococcal CAP were detected solely through UAD-24 testing.

The virulence of *S. pneumoniae* serotypes has been assessed based on their ability to cause invasive disease. Serotypes isolated from blood or other sterile sites are categorized as invasive serotypes and are generally regarded as the most virulent. However, this link between serotype invasiveness and virulence has been questioned by some investigators, as non-invasive serotypes have also been associated with poor clinical outcomes [13]. In patients with pneumococcal pneumonia, serotype virulence may be better characterized by those serotypes associated with patients with severe clinical presentations and poor clinical outcomes. The association of non-invasive serotypes and clinical outcomes is facilitated by the use of UAD-24 technology. Since the host immune response plays a critical role in determining the outcomes of patients with pneumococcal pneumonia, a comprehensive assessment of *S. pneumoniae* serotypes should consider the potential association of serotypes in patients with compromised immune responses due to underlying medical conditions or medical treatments.

Considering the above information, we designed a study using UAD-24 technology with the objectives to (1) define the incidence, epidemiology, and mortality of adult patients hospitalized with pneumococcal pneumonia in Louisville, Kentucky, (2) estimate the burden of pneumococcal pneumonia in the US adult population, and (3) characterize the association of *S. pneumoniae* serotypes with patients’ comorbidities and outcomes.

## 2. Materials and Methods

### 2.1. Study Design and Study Population

This was a nested study of a prospective population-based cohort study of all adult residents in Louisville, who were hospitalized with CAP from 1 June 2014 to 31 May 2016 [14]. After informed consent was obtained, a urine sample was collected and a UAD-24 test was performed by capturing pneumococcal polysaccharides with serotype-specific monoclonal antibodies using Luminex technology [11,12]. For consenting patients, Quellung reaction was used to identify serotypes of *S. pneumoniae* isolated from the blood and respiratory samples obtained during standard of care.

### 2.2. Inclusion and Exclusion Criteria

Adult patients aged 18 years or older who were hospitalized with CAP were eligible for inclusion in this study. A patient was defined as having CAP when the following three criteria were met: (1) the presence of a new pulmonary infiltrate on chest radiograph and/or chest computed tomography scan at the time of hospitalization; (2) the presence of at least one of the following: (a) new cough or increased cough or sputum production, (b) fever > 37.8 °C (100.0 °F) or hypothermia < 35.6 °C (96.0 °F), and (c) changes in leukocyte count (leukocytosis: >11,000 cells/mm^3^; left shift: >10% band forms/mL; or leukopenia: <4000 cells/mm^3^); and (3) no alternative diagnosis at the time of hospital discharge that justified the presence of criteria 1 and 2.

With the intent to enroll only hospitalized patients with CAP who lived in Louisville, Kentucky, and who were counted in the 2010 US Census, patients were excluded from analysis if they (1) did not have a permanent or valid Louisville address based on the US Census Bureau data, (2) did not have a valid Social Security Number (SSN), or (3) were in the correctional system.

### 2.3. Human Subject Protection

This nested sub-study and the parent study were approved by the Institutional Review Board at the University of Louisville Human Subjects Research Protection Program Office (IRB # 11.0613, # 13.0408) and by the research offices at each participating hospital.

### 2.4. Specimen Collection and Definition of Pneumococcal Pneumonia

Specimens for *S. pneumoniae* testing were collected after informed consent was obtained. A patient was defined as having pneumococcal pneumonia when any of the following three criteria were met: (1) positive culture of *S. pneumoniae* from respiratory or blood samples; (2) positive urinary antigen detection of *S. pneumoniae* polysaccharide C; or (3) positive urinary antigen detection of any 24 *S. pneumoniae* serotypes via the UAD-24 test.

### 2.5. Incidence Calculation in Louisville

Among the patients with a UAD-24 test performed, the number of pneumococcal pneumonia cases was defined. The proportion of pneumococcal pneumonia cases was the number of pneumococcal pneumonia cases divided by the number of patients in whom a UAD-24 test was performed. This proportion was used as a multiplier to estimate the number of pneumococcal pneumonia cases in patients without a UAD-24 test performed. Wilson 95% confidence intervals were calculated for the number of estimated cases. Annual cumulative incidence of pneumococcal pneumonia hospitalizations per 100,000 adults was calculated using the estimated number of pneumococcal pneumonia cases divided by the number of adults living in Louisville according to the US census data [15]. Complete descriptions of the incidence calculations can be found in the Appendix A. R software (version 4.2.0, R Core Team, Vienna, Austria) was used for all statistical analysis.

### 2.6. Geospatial Epidemiology

The geomasked location of each patient’s home address who was diagnosed with pneumococcal pneumonia was obtained through the US Census Bureau website [16]. A kernel density heat map was created using each patient’s home location at the time of hospitalization during the study period. Kulldorff’s Spatial Scan Statistic was used to determine significant areas of risk for hospitalization due to pneumococcal pneumonia [17]. Using census tract-level demographics, kernel density heat maps were created for (1) the Louisville population living in poverty, (2) the population that identifies as Black or African American race, and (3) the population that is aged 65 years or older. These maps were compared to the spatial distribution of pneumococcal pneumonia cases. A complete description of the geospatial methods can be found in the Appendix A. ArcGIS (version 10.7, ESRI, Redlands, CA, USA) was used for all geospatial analyses.

### 2.7. Clinical Outcomes

At the time of hospital admission, patients were considered as having severe CAP if they required direct admission to ICU. Patients were also classified according to the Pneumonia Severity Index (PSI) risk class [18]. Clinical failure in patients was defined as evidence of any of the following during hospitalization: (1) acute pulmonary deterioration with the need for mechanical ventilation or (2) acute hemodynamic deterioration with the need for vasopressors [19]. A patient was defined as having a cardiovascular event if any of the following events were diagnosed during hospitalization: myocardial infarction, new-onset cardiac arrhythmia, acute worsening of a long-term arrhythmia, cerebrovascular accident, pulmonary embolism, or pulmonary edema.

All-cause mortality for hospitalized patients with pneumococcal pneumonia was evaluated during hospitalization and at 30 days, 6 months, and 1 year after hospitalization.

### 2.8. Extrapolations to the United States

The number of pneumococcal pneumonia hospitalizations in the US was estimated by multiplying the Louisville cumulative incidence rate by the estimated 2014 US adult population obtained from the US Census Bureau [15]. Complete descriptions of the incidence calculations can be found in the Appendix A.

Mortality data from Louisville were extrapolated to the US population hospitalized with pneumococcal pneumonia. Full methodologic descriptions can be found in the Appendix A.

### 2.9. Serotype Incidence

For patients with a UAD-24 test performed, the incidence of *S. pneumonia* serotypes was defined as the proportion of *S. pneumoniae* serotypes from all detected serotypes. This was calculated as the percentage of each serotype from the total number of serotypes identified from any sample source.

### 2.10. Serotype Associations with Comorbidities and Clinical Outcomes

For each serotype detected by UAD-24, we considered the total number of said serotype detected to be 100%. Based on this 100% per serotype, the prevalence of clinical characteristics and outcomes was calculated for each serotype. The clinical characteristics evaluated per serotype were patient comorbidities. The outcomes evaluated per serotype were severity of disease, clinical failure, cardiovascular events, and in-hospital mortality. Serotypes were excluded from this analysis if there were less than 6 patients with the serotype detected.

### 2.11. Serotype Virulence

For each clinical outcome, serotypes were ranked according to their prevalence, so that the serotypes with the highest prevalence received the highest rank. Using these ranks, the average rank of all outcomes for each serotype was calculated. The serotypes with the highest average rank of these clinical outcomes were considered the most virulent.

## 3. Results

### 3.1. Incidence of Pneumococcal Pneumonia

#### 3.1.1. Proportion of Pneumococcal Pneumonia

Of the 8284 patients hospitalized with CAP in the parent study, 6196 patients consented for urinary antigen testing. Among these patients, 5402 had samples for urinary antigen detection of 24 *S. pneumoniae* serotypes and Quellung reaction for identifying *S. pneumoniae* obtained during standard of care. These 5402 patients constituted the study population. A flowchart describing the study participants is depicted in Figure 1. From the study population, pneumococcal pneumonia was identified in 708 patients (13%). The characteristics of patients with pneumococcal pneumonia are depicted in Table 1. A detailed breakdown of the identification of *S. pneumoniae* by sample type is depicted in the Appendix A. The proportion of pneumococcal pneumonia cases was determined to be 13% based on the number of patients with *S. pneumoniae* identified in the study population.

#### 3.1.2. Incidence of Hospitalizations Due to Pneumococcal Pneumonia

To calculate the incidence of pneumococcal pneumonia, the *S. pneumoniae* multiplier was first used to estimate the number of *S. pneumoniae* cases detected in patients without a UAD-24 test performed. From the 2882 patients, and using the 13% multiplier, 378 *S. pneumoniae* CAP cases were estimated. This leads to 1081 cases of *S. pneumoniae* CAP over the two-year period, and 541 cases of *S. pneumoniae* CAP annually. Using the adult population of Louisville, KY, the annual incidence of hospitalizations due to pneumococcal pneumonia was 93 cases per 100,000 adults (95% CI = 91–95).

#### 3.1.3. Incidence of Pneumococcal Pneumonia by Age and Comorbidity

Figure 2 depicts the incidence of pneumococcal pneumonia by age group. Older adult patients had a higher incidence of hospitalization due to pneumococcal pneumonia. Figure 3 depicts the incidence of pneumococcal pneumonia by comorbid condition. A history of COPD was the comorbid condition associated with the highest incidence of hospitalization due to pneumococcal pneumonia.

#### 3.1.4. Association of Pneumococcal Pneumonia with Income Level, Race, and Age

The kernel density heat map of unique patients with pneumococcal pneumonia in the city of Louisville is depicted in Figure 4A. Accounting for the underlying population density, a zone of high risk for hospitalization due to pneumococcal pneumonia was identified in the western part of the city (Risk Ratio: 2.13, *p* < 0.001). The clustering of *S. pneumoniae* cases in the western part of the city was associated with census tracts where the average population had a larger proportion of individuals with a low annual income (Figure 4B) or a high proportion of individuals of Black or African American race (Figure 4C). Census tracts with the highest percentage of older adult population were located in the eastern part of the city (Figure 4D).

### 3.2. Clinical Outcomes

#### 3.2.1. Severe CAP, Clinical Failure, and Cardiovascular Events

A total of 407 (57%) patients with pneumococcal pneumonia had PSI risk class IV or V at the time of hospitalization, and a total of 144 (20%) patients with pneumococcal pneumonia required ICU admission at the time of hospitalization. Clinical failure was documented in 147 (21%) patients. Cardiovascular events, documented in 55 (8%) patients, were as follows: new arrhythmia or worsening of a long-term arrhythmia in 45 (6%) patients, acute myocardial infarction in 11 (2%) patients, pulmonary embolisms in 2 (<1%) patients, and pulmonary edema in 2 (<1%) patients.

#### 3.2.2. Mortality

The all-cause mortality for the 708 hospitalized adult patients with pneumococcal pneumonia in the city of Louisville was as follows: 3.7% during hospitalization, 8.2% at 30 days, 17.6% at 6 months, and 25.4% at 1 year after hospitalization.

### 3.3. Estimation of Burden of Disease in the United States

Using the incidence of pneumococcal pneumonia, the estimated number of hospitalizations due to pneumococcal pneumonia in the US was 226,696 per year. Using the Louisville mortality data, the number of estimated deaths in the US population was 8325 during hospitalization, 18,624 at 30 days, 39,817 at 6 months, and 57,641 at 1 year after hospitalization.

### 3.4. Serotypes

#### 3.4.1. Serotype Incidence

Among the 708 patients with pneumococcal pneumonia, 519 patients had serotypes identified from any sample type, with 581 serotypes identified in total. The serotype distribution for patients in whom a serotype was identified is shown in Figure 5.

#### 3.4.2. Serotype Associations with Comorbidities and Clinical Outcomes

*S. pneumoniae* serotypes associated with the highest rates of neoplastic disease, chronic obstructive pulmonary disease, diabetes, renal disease, cardiovascular disease, and liver disease are shown in Figure 6. *S. pneumoniae* serotypes associated with the highest rates of PSI risk class IV/V, ICU admission, clinical failure, cardiovascular events, and in-hospital mortality are shown in Figure 7.

Serotypes 23F, 1, and 3 were the highest ranked serotypes in terms of clinical outcomes. The average rank of serotype virulence is depicted in Figure 8.

## 4. Discussion

This study indicates that in Louisville, the annual incidence of adults hospitalized with CAP due to *S. pneumoniae* is 93 per 100,000 adults (95% CI = 91–95). This translates to an estimated 225,000 adult hospitalizations in the US annually. Adults aged 65 years or older have a substantially higher rate of hospitalization due to pneumococcal pneumonia compared to adults aged 18–64 years old (280 per 100,000 versus 53 per 100,000, respectively). Several comorbid conditions place patients at an increased risk for pneumococcal pneumonia. In our study, patients with COPD have the highest risk for hospitalization due to *S. pneumoniae* CAP, with an annual incidence of 982 per 100,000 adults with COPD.

The extrapolations from the city of Louisville to the United States are based on our prior work indicating that the city of Louisville is a microcosm of the United States. We have previously evaluated 52 demographic, socioeconomic, and health behavior variables of cities in the United States to define how well they represent the overall United States census demographics [20]. From all cities in the US with a population greater than 500,000, we identified Louisville as the second closest city to the overall United States [20]. This dataset provides support for our national estimation of burden of pneumococcal pneumonia. Of note, the state of Kentucky has significantly more comorbid cases and a higher incidence of tobacco use, including smoking, than Louisville and the overall United States.

In our study, we identified that 13% of adult patients hospitalized with CAP have pneumococcal pneumonia. This rate is significantly lower than the rates reported in two European studies of adults hospitalized with CAP that utilized UAD-24 testing, which found 24–29% of patients had pneumococcal pneumonia [7,8]. Our findings agree with the current literature indicating that the incidence of pneumococcal CAP is less prevalent in the United States compared to Europe [21]. In the US, surveillance studies of pneumococcal pneumonia have been limited to invasive serotypes [22]. Further studies using UAD-24 technology will be necessary to define current trends of non-invasive pneumococcal pneumonia in the US.

Geospatial epidemiology indicates there is an increased risk for pneumococcal pneumonia in the western half of the city. This corresponds to areas of the city that have both a high prevalence of poverty and a high proportion of Black or African American residents. This ecological finding is in accordance with a prior CDC study evaluating socioeconomic and racial disparities in 4870 adults with bacteremic pneumococcal pneumonia [23]. Since there is no genetic predisposition for contracting pneumococcal pneumonia among Black or African American individuals, the geospatial association of race and pneumococcal pneumonia is most likely explained by poverty and other determinants of health. Poverty is often recognized as a marker for several factors that may increase the risk of pneumococcal pneumonia, such as poor nutrition, increased number of occupants per room, poor air quality, increased rates of smoking, lack of medical insurance, and suboptimal access to medical care and preventive medicine.

In the current study, we evaluated the severity of pneumococcal pneumonia at the time of hospitalization and clinical outcomes. Regarding severity of disease, over half of the patients hospitalized with pneumococcal pneumonia had PSI risk class IV or V, and one in five required ICU admission at the time of hospitalization. After hospitalization, nearly one in five patients developed clinical failure, requiring mechanical ventilation or vasopressors. In agreement with the current literature, we found that 8% of patients with pneumococcal pneumonia developed cardiovascular events during hospitalization, the most common of which was new onset of arrhythmia [24].

Our findings show that approximately one in four patients hospitalized with pneumococcal pneumonia will die within one year after hospitalization. Annually, we estimated that approximately 57,000 patients in the US will die within one year after hospitalization for pneumococcal pneumonia. Excess death after hospital discharge for patients admitted with CAP when compared to control subjects has been previously reported [25,26,27,28]. Systemic inflammation has been documented in patients with pneumococcal pneumonia after a clinical resolution of the infection. Persistent inflammation after an episode of pneumococcal pneumonia may play a role in the premature death observed in these patients.

The most common serotypes detected in our study were 19A, 3, and 22F, accounting for nearly one third of all identified pneumococcal serotypes detected. All these serotypes are present in currently recommended conjugated vaccines [29]. These three serotypes were also commonly found in a combined total of almost 1200 patients with pneumococcal pneumonia in three studies utilizing UAD-24 serotype testing in Europe [6,7,8]. The most significant difference between these European studies and our US data is the high frequency of serotype 8 in each European study and the low frequency of serotype 8 in our US data [6,7,8].

We attempted to define which *S. pneumoniae* serotypes were most virulent based on their association with more severe clinical diseases and worse clinical outcomes. Based on the average rank of the outcomes, serotypes 23F, 1, and 3 were the most virulent serotypes, as patients hospitalized with these three serotypes experienced the highest rates of severe CAP and poor clinical outcomes compared to other serotypes. Our findings agree with a study in Spain utilizing UAD-24 testing, which found serotype 3 to be most associated with severity of disease and serotypes 1 and 3 to be most associated with poor outcomes [8]. One major difference was the identification of serotype 23F as a highly virulent serotype in our data [8]. We did not find any specific trends in serotype associations with patients’ comorbidities.

One limitation of our study is that serotyping was primarily performed using the UAD-24 platform. Considering that there are 100 known serotypes of *S. pneumoniae* [30], we might have underestimated the true incidence of *S. pneumoniae* as the etiology of CAP in hospitalized patients, as well as the distribution of serotypes among pneumococcal pneumonia cases. As UAD-24 testing is a non-commercial test, its availability is limited and its clinical use has not been fully evaluated. Another limitation is that among the 8284 patients with CAP, we were able to perform UAD-24 testing in only 5402 patients. Finally, this study was performed in the pre-COVID-19 pandemic period. In the current post-pandemic period, the incidence and epidemiology of *S. pneumoniae* may have been altered.

Our study has several strengths. First, we were able to evaluate and enroll all adult hospitalizations in the city of Louisville for two consecutive years. Second, we were able to identify cases of pneumococcal pneumonia that were also included in our defined geographic area through the US Census Bureau, using the patients’ home addresses. Third, we were able to define the number of unique patients hospitalized with pneumococcal pneumonia using the patients’ Social Security Number.

## 5. Conclusions

In conclusion, we documented an annual incidence of 93 adults hospitalized with pneumococcal pneumonia per 100,000 adults in the city of Louisville. We estimated a substantial burden of pneumococcal pneumonia in the US adult population. Approximately 225,000 unique adults will be hospitalized in the US each year due to pneumococcal pneumonia. One year after hospitalization due to pneumococcal pneumonia, death will occur in nearly one out of four adults. Due to this excess burden of disease, efforts to advance prevention strategies and treatment modalities are needed.

## Figures and Tables

**Figure 1 microorganisms-11-02813-f001:**
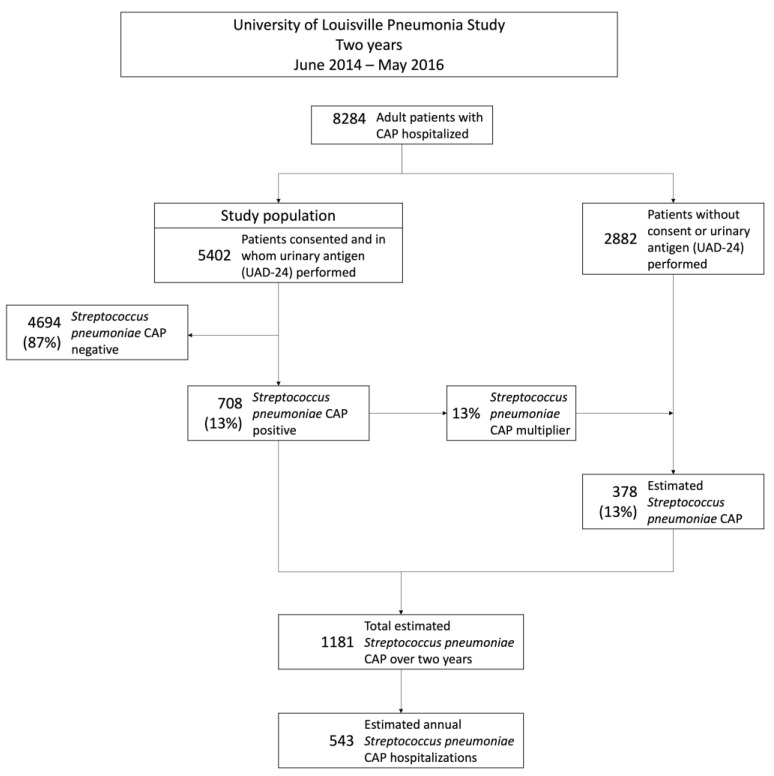
Flow chart of study participants with estimation of the annual number of adults hospitalized with *S. pneumoniae* CAP.

**Figure 2 microorganisms-11-02813-f002:**
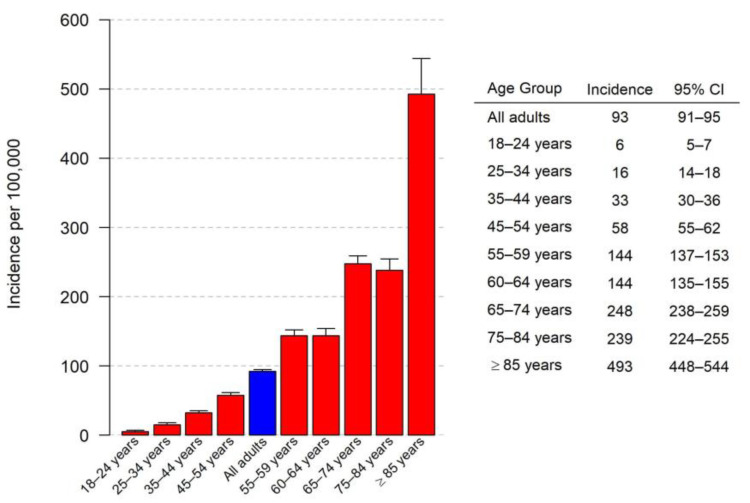
Incidence of *S. pneumoniae* CAP per 100,000 adults by age group.

**Figure 3 microorganisms-11-02813-f003:**
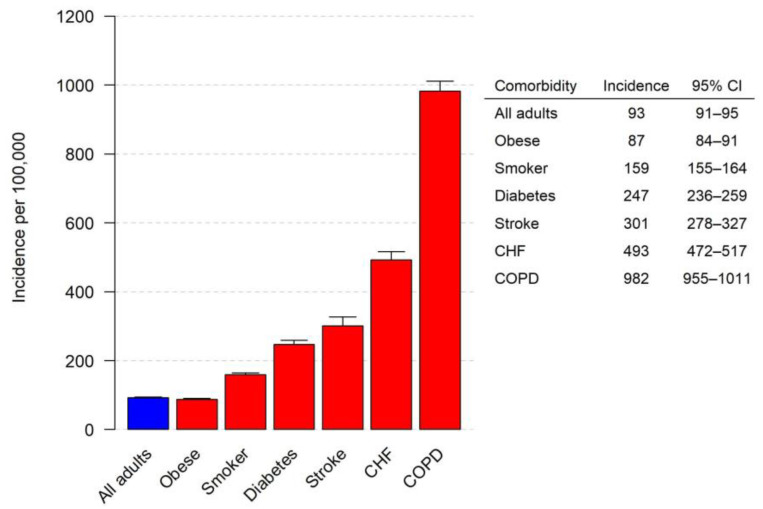
Incidence of *S. pneumoniae* CAP per 100,000 adults by comorbidity.

**Figure 4 microorganisms-11-02813-f004:**
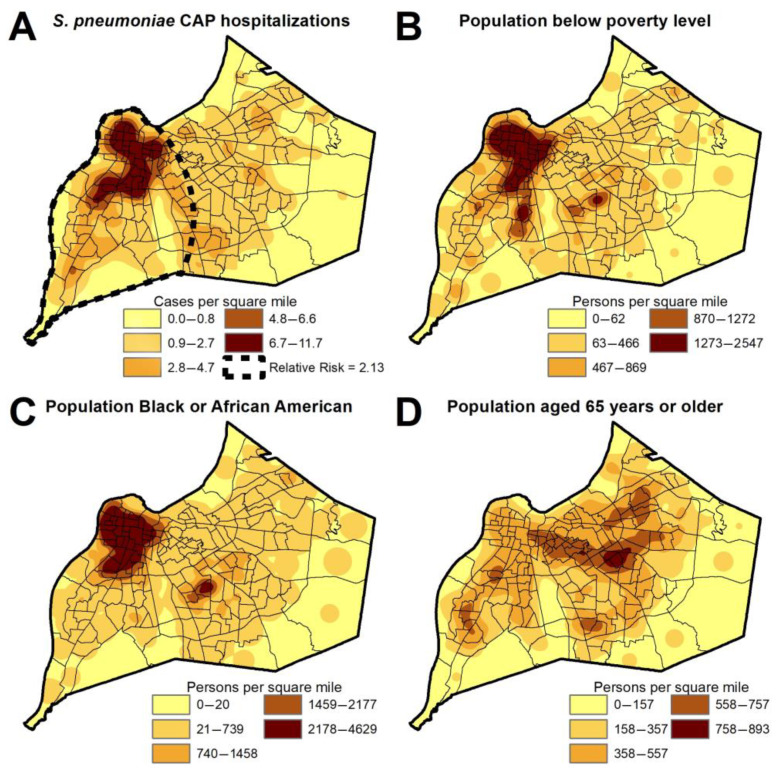
Heat maps of the city of Louisville for patients hospitalized with *S. pneumoniae* CAP (**A**), population living in poverty (**B**), population that identifies as Black or African American race (**C**), and population that is aged 65 years or older (**D**). The area of relative risk for hospitalization with *S. pneumoniae* CAP in the west part of the city is depicted by the dotted line in map (**A**).

**Figure 5 microorganisms-11-02813-f005:**
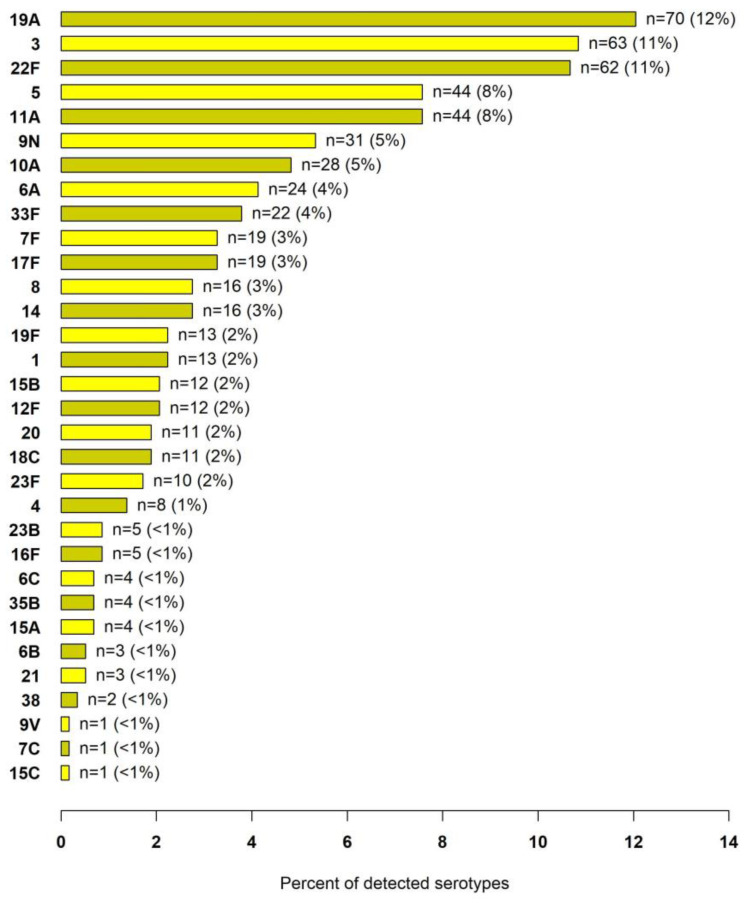
Distribution of the 581 *S. pneumoniae* serotypes detected from any sample type.

**Figure 6 microorganisms-11-02813-f006:**
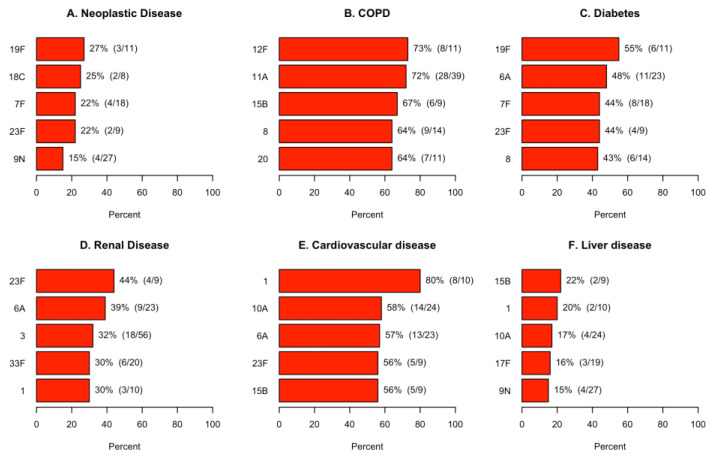
Estimation of the most associated serotypes found by UAD-24 testing by comorbidity. Only the top five serotypes are depicted for each comorbidity.

**Figure 7 microorganisms-11-02813-f007:**
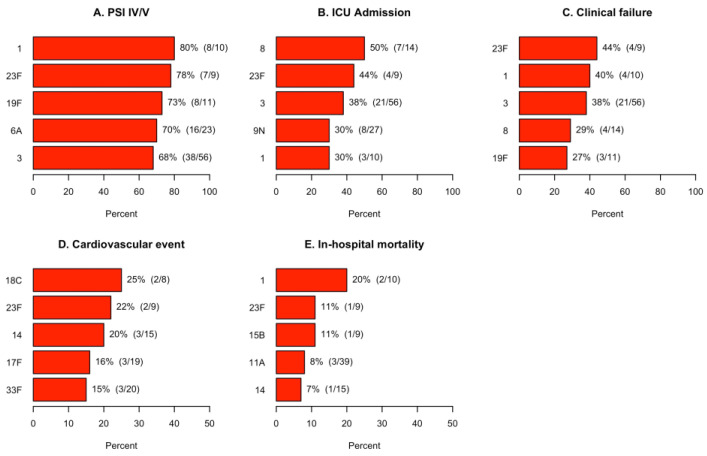
Estimation of most associated serotypes found by UAD-24 testing with severe CAP, clinical failure, cardiovascular events, and in-hospital mortality. Only the top five serotypes are depicted for each outcome.

**Figure 8 microorganisms-11-02813-f008:**
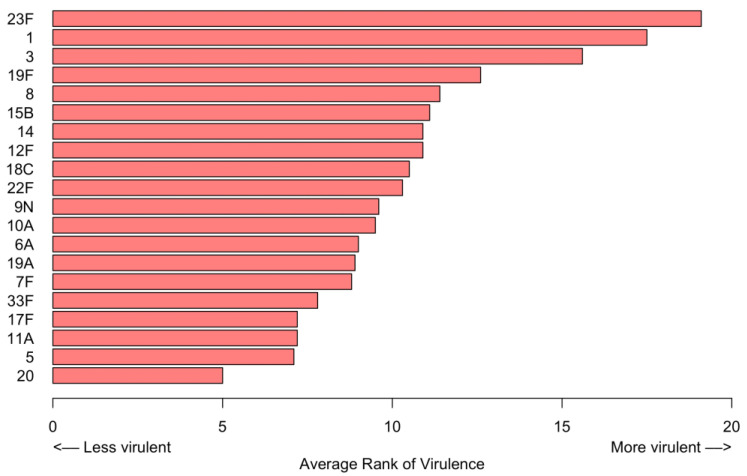
Estimation of virulence based on the average rank of outcomes. A higher rank indicates more virulence.

**Table 1 microorganisms-11-02813-t001:** Characteristics of hospitalized patients with *S. pneumoniae* CAP (n = 708).

Variable	Value
Total N	708
Demographics	
Age, median (IQR)	65 (56–75)
Male sex, n (%)	323 (46)
Black or African American race, n (%)	123 (17)
Medical and social history, n (%)	
Obese	238 (34)
HIV disease	21 (3)
Neoplastic disease (active or within the last year)	86 (12)
Renal disease	184 (26)
Chronic renal failure	44 (6)
Congestive heart failure (CHF)	199 (28)
Chronic obstructive pulmonary disease (COPD)	403 (57)
Stroke	85 (12)
Current smoker	317 (45)
Diabetes mellitus	214 (30)
Severity of disease	
ICU admission, n (%)	144 (20)
Altered mental status on admission, n (%)	96 (14)
Vasopressors on admission, n (%)	25 (4)
Ventilatory support on admission, n (%)	100 (14)
Pneumonia Severity Index, median (IQR)	97 (74–127)
PSI risk class IV or V, n (%)	407 (57)

## Data Availability

The data presented in this study are available from the corresponding author upon request. The data are not publicly available due to the privacy of the participants.

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
