# Peer review of "Epidemiology of Pneumococcal Pneumonia in Louisville, Kentucky, and Its Estimated Burden of Disease in the United States"

_microorganisms, 2023, doi:10.3390/microorganisms11112813_

Round 1

Reviewer 1 Report

Comments and Suggestions for Authors

The research article of Ramirez et al. is focused on the epidemiological aspects of pneumococcal pneumonia in Kentacky during 2 years, 2014-2016. Paper is writen clearly, I have only few minor comment:

- I like the short introduction, but in this case, it can be a little prolongated - what is missing here from my point of view is - epidemiology in other countries (morbidity and mortality in USA/worldwide, available detection methods (only UAD is mentioned), limitations of UAD.

-Also, in the discussion, I am missing older data, to get a overview of the trend of pneumococcal pneumona in the region (they can be find in older papers).

-By seaching about the topics I found that in Kentacky is much higher death rate of pneumonia compare to the rest of the USA - quite surprising, maybe autors can answer, why (it is only the personal interest, not necessary include this to the review).

-Material and methods - please add a number of samples (patients)

- please change to italics scientific names (L16, L199, L206, L215, L263...)

-L51 - the (12) is a citation? It has a different format from other citations. Same for the citations(?) in the discussion, they are differently formated

Reviewer 2 Report

Comments and Suggestions for Authors

Dear authors,

Worldwide, lower respiratory tract infections are a major cause of morbidity and mortality. Streptococcus pneumoniae remains indeed a primary pathogen in hospitalized patients with community-acquired pneumonia. For this reasons I think that this article is very important.

 Came back to the text:

- the introduction is to short, must to extend

- Chronic renal failure (CHF)- is communly CRF, or AKI, CHF is hepatic failure. What did you want to say?

- the method and result are corectelly described

- maybe can move the figure 5 and 6 from the discussion section to the result.

- the discussion are a little to short. By extension of this section, you can improve the references also.

You can discuss about the incidence of some serotypes in the high risk patients, for example the old patients, the patients with diabetes, cardiopathies, CKD or immunodeficiency.
